# THE ART OF BREAKING WORDS:
# RETHINKING MULTILINGUAL TOKENIZER DESIGN

## ABSTRACT

While model architecture and training objectives are well-studied for Large Language Model (LLM), tokenizer, particularly in multilingual contexts, remains a relatively neglected aspect of LLM development. Existing multilingual tokenizers often exhibit high token-to-word ratios, leading to inefficient use of context length and slower inference. This motivated us to conduct a systematic study that links vocabulary size, pre-tokenization rules, and training-corpus composition to both token-to-word efficiency and model quality. To ground our analysis in a linguistically diverse context, we conduct extensive experiments on Indic scripts, which present unique challenges due to their high script diversity and orthographic complexity. Drawing on the insights from these analyses, we propose a novel algorithm **AdaptMix**, for data composition that balances multilingual data for tokenizer training.

Our observations on pre-tokenization strategies significantly improve model performance, and our data composition algorithm (AdaptMix) reduces the average token-to-word ratio by approximately 6% with respect to the conventional data randomization approach. Our tokenizer achieves more than 40% improvement on average token-to-word ratio against state-of-the-art multilingual Indic models. This improvement yields measurable gains in both model performance and inference speed. This highlights tokenization alongside architecture and training objectives as a critical lever for building efficient, scalable multilingual LLMs.

## 1 INTRODUCTION

In the domain of natural language processing(NLP) tokenizer is a fundamental component to bridge human-readable text and model-readable tokens. Tokenizer algorithm such as WordPiece Devlin et al. (2019), BPE Sennrich et al. (2016), Unigram Kudo & Richardson (2018) and Fast Word-Piece Song et al. (2021) form the basis of modern NLP. Tokenizer can significantly influences the efficiency, inference speed and context length of deep-learning models especially for transformers Vaswani et al. (2017a). Indian languages are characterized by their linguistic diversity and multiple scripts, including native scripts such as Devanagari and Dravidian, as well as transliterated forms in Latin scripts. Native scripts dominate formal contexts such as literature, and academic publications, whereas informal and digital communication increasingly employ Latin scripts.

Existing multilingual models like Bloom Luccioni et al. (2023); Muennighoff et al. (2023), LLaMA Touvron et al. (2023a;b); Grattafiori et al. (2024), Gemma3 Team et al. (2024a;b; 2025), Mistral Jiang et al. (2023), Qwen Bai et al. (2023); Yang et al. (2024); Qwen et al. (2025); et al. (2025), Nemotron Nvidia et al. (2024), Sarvam AI (2024), Param Pundalik et al. (2025) often demonstrate suboptimal performance on Indian languages due to their predominantly Latin-centric vocabularies. Consequently there is a pressing need for tokenizer strategies that efficiently handle both native and transliterated scripts to accommodate the prevalent code-mixing and the multilingual nature of Indian digital communication.

Addressing these challenges, we conduct an extensive study on various pre-tokenization strategies and a novel adaptive data mixture algorithm(**AdaptMix**) for training a multilingual tokenizer. Method leverages multilingual datasets to dynamically balance language representation, considerably improving tokenization quality. Empirical results demonstrate our algorithm achieves sig-

nificant improvement in token-to-word ratio compared to standard baselines, enhancing tokenizer performance. Increasing inference speed of model and efficient context length usage of model.

## 2 RELATED WORK

Tokenizer has evolved significantly over the past years, particularly with the adoption of subword tokenizer algorithms like Byte Pair Encoding (BPE) Sennrich et al. (2016), Unigram language models Kudo (2018), and SentencePiece Kudo & Richardson (2018). Further advances in word representation were achived by Radford et al. (2019), which introduced byte-level tokenizer, and by Provilkov et al. (2020), which proposed BPE-dropout.

While an increasing amount of research explores tokenizer development, most existing studies only provide high-level descriptions of their approaches and rarely disclose detailed empirical data distributions influencing vocabulary design. There are a few notable exceptions though, such as Dagan et al. (2024a); Reddy et al. (2025), that present extensive empirical analysis on the size of the training data for tokenizers. Several notable approaches like MorphTok Brahma et al. (2025) introduces manually curated word-set and architectural changes for Indic language, however these methods are time-consuming to implement and challenging to generalize across languages.

However, in multilingual settings, the data mixture used to train tokenizers is critical but often overlooked. Models such as XLM-R Lample & Conneau (2019) and mT5 Xue et al. (2021b) typically construct their vocabulary using large multilingual corpora where data is sampled in proportion to language availability. This sampling strategy can disproportionally favor high resource languages, resulting in lower tokenization efficiency for low resource morphologically complex languages. Even though byte and character level models such as ByT5 Xue et al. (2021a) and Canine Clark et al. (2021) mitigate this by operating below the subword level, they introduce significantly longer sequences and hence, increased computational costs.

Despite substantial progress in tokenization techniques, a key gap remains in how multilingual data is composed during vocabulary construction. This calls for a more careful consideration of data mixture strategies that go beyond corpus size and incorporate linguistic and structural diversity to increase the efficiency of tokenization across languages.

## 3 METHOD

The primary objective is to design and implement a tokenizer that can effectively process diverse Indic linguistic styles. This includes support for all 22 officially recognized Indian languages and widely used programming languages that require precise syntactic parsing.

We adopt SentencePiece Kudo & Richardson (2018) algorithm, for training our tokenizer due to its effectiveness in handling diverse scripts. The datasets span multiple categories such as synthetic corpora, scraped text, code and mathematical corpora further explained in 3.1. We perform multiple experiments on different vocabulary size to get optimal size for multilingual Indian languages, further described in 3.2. Extensive experimentation is done to identify suitable pre-tokenization strategies for Indic languages. To optimize the tokenizer performance across multiple languages and domains, we used our novel algorithm 3.4 and compared with state-of-the-art tokenizers in 4.3.

### 3.1 DATASET

To build our tokenizer, we curated a diverse multilingual and multi-domain dataset spanning 16 Indian languages (native and Latin scripts), programming languages, and LaTeX content. Sources include open corpora, web-scraped and OCR data, and synthetic examples.

**Open-Source Dataset:** We have included, more than 35 open source datasets, including Sangraha Khan et al. (2024), Samanantar Ramesh et al. (2022), NLLB Team et al. (2022), Wikilingua Ladhak et al. (2020), the Pile Gao et al. (2021), and IndicCorp Kakwani et al. (2020). Additionally, raw data covering 16 Indic languages was scraped from web sources and parsed through the following steps: (1) Boilerplate and HTML Removal, (2) Unicode Normalization, (3) Repetition and Noise Removal, (4) Global Deduplication, (5) Language and Length Filtering. Prior to sampling, the corpus was classified into different quality using in house quality classification pipelines.

Only high-quality segments from each dataset were retained. The selected data was shuffled and randomly sampled to ensure broad domain coverage and vocabulary diversity across languages.

**Synthetically Curated Data** Despite India's linguistic diversity, many Indic languages, including Maithili and Sindhi (Devanagri), remain severely underrepresented in publicly available corpora. Web scraped data is disproportionally skewed towards English and a few high resource languages like Hindi, leaving limited high quality data for effectively training models and tokenizers.

To address this, we utilized a large scale synthetic indic corpus using persona driven generation Ge et al. (2025). Drawing upon over 100 million Indian personas across 16 domains and 100+ fine grained roles contributed to the development of synthetic data for our tokenizer training. Outputs were generated using open source and filtered for quality, and averaged 900-1000 words per sample. To enhance Indic language coverage, these English passages were translated into 15+ Indian languages using a 2 stage neural machine translation pipeline. Initial translation was performed using IndicTrans2 Gala et al. (2023), followed by post correction through open source LLMs.

## 3.2 VOCABULARY

To determine the optimal vocabulary size capable of supporting diverse linguistic and structural complexity present in Indic languages, programming syntax, and mathematical notations, an extensive series of ablation studies was conducted. These studies aimed to evaluate the effects of different vocabulary sizes on the tokenization granularity by analyzing token-to-word ratio – defined as the average number of tokens generated per word.

To ensure comprehensive language coverage, we explicitly incorporated all unique characters across Indian languages in the vocabulary prior to the tokenizer training. Due to extensive character set inherent in Indic scripts, this approach prevents the over-fragmentation of rarely occurring characters which is not present in training dataset. Moreover, the vocabulary includes special tokens such as pad, start and end of the sentence, as well as multiple instruction tokens like tools, user and assistant intended for fine-tuning the model in the downstream task. To accommodate future expansion, multiple tokens are intentionally left unassigned, providing flexibility for domain-specific adaptation.

## 3.3 PRE-TOKENIZATION

Pre-tokenization rules are vital for building efficient tokenizer, as they standardize input text and reduce redundancy. It ensures that words with minor diacritic variations are correctly distinguished. Effective pre-tokenization enables the model to learn representation efficiently and optimize vocabulary usage, since entities with the same sub-word mostly has similar semantic meanings.

Individual digits, including Indic scripts, are also split during pre-tokenization to support the generalization of basic arithmetic or logical reasoning. Prior studies Nogueira et al. (2021), Thawani et al. (2021), Dagan et al. (2024b) have shown that splitting digits can positively impact the performance of arithmetic tasks. Similarly, splits are performed on line breaks and trailing whitespace. Taking programming formats into consideration to prevent long context lengths due to these splits, multiple groups of whitespace are implicitly added.

We experimented with pre-tokenization strategies, with multiple methods of diacritic separation. This approach considers a trade-off between token-to-word ratio and the model's linguistic comprehension. Indic scripts, being largely phonetic are prone to errors, especially writing diacritics by the end-user, which can significantly distort embedding representation during inference. While a large portion of training data is either synthetically generated or carefully written and thus free from these types of errors, these representation won't be learned by the model and hence might be unable to provide response to end-users correctly. Moreover these errors will also increase the token-to-word ratio. These discrepancies alter the token embeddings and can impact model performance. By applying pre-tokenization, we believe the model's complexity in handling these variations can be reduced. Two pre-tokenization strategies were evaluated: one involving the separation of all diacritics, and another separating only a subset to optimize the token-to-word ratio. These were compared against a baseline tokenizer with no pre-tokenization, along with corresponding models trained using each tokenizer variant.

## 3.4 ADAPTMIX: ADAPTIVE DATA MIXTURE

In earlier experiments, we consistently observed that languages with high token-to-word ratio such as Sanskrit often exhibit morphological richness and orthographic complexity. Morphologically rich languages encode grammatical meaning through extensive inflection and compounding, resulting in long and variable word forms. Similarly, scripts such as Malayalam and Devanagari include ligatures, diacritics, and non-linear character arrangements, increasing the likelihood of token fragmentation. This suggested that uniform sampling ignores the linguistic complexity of each language, causing inherently harder languages to under perform even when equally represented. While there has been growing interest in optimizing data mixtures for pretraining large language models, such as in works like DoReMi Xie et al. (2023) and DRO Oren et al. (2019), similar exploration for tokenizer training remains limited, especially in multilingual contexts. Some prior efforts, such as the approach used in Lample & Conneau (2019), attempt to mitigate resource imbalance during training by sampling sentences from each language according to a smoothed distribution, but the sampling remains fundamentally tied to corpus availability.

To address these limitations, we propose an adaptive mixture strategy that dynamically adjusts language wise sampling proportions based on current token-to-word ratio. This improves representation of under performing languages and gradually steer the tokenizer training towards a balanced state, where improvements in one language no longer come at a significant cost to others.

Higher token-to-word ratio indicates that the tokenizer fragments words into more units, which reflects low tokenization efficiency. Our algorithm incorporates an iterative feedback loop of training tokenizers, allowing the mixture to adapt over time towards a balanced configuration. This feedback-driven optimization progressively reallocates training data in response to observed inefficiencies in token-to-word ratio, aiming to reach an equilibrium.

For each language $i \in L$, a *scaled token-to-word ratio* $f_i^n$, also known as fertility, is computed to quantify tokenization inefficiencies in language relative to the target fertility (kept constant at 1), and normalized by the fertility range. If languages happen to have the same token-to-word ratio, the optimizer simply reuses the previous mixture proportions, rescaled to match the sampling budget.

$$\delta_l^N = \frac{f_i^N - f_{\text{best}}}{f_{\text{range}}^N} \tag{1}$$

As lower values of token-to-word ratio or fertility are preferred, $f_{best} = \min_{i \in L} f_i^N$. A small constant is added to each $\delta_l^N$ to ensure all languages retain non-zero weight, preventing exclusion.

$$w_l^N = \delta_l^N + \varepsilon \tag{2}$$

The resulting deficit weights are normalized across all languages to ensure that resulting values form a valid probability distribution. These proportions reflect the composition for next traning iteration, based purely on its relative tokenization performance. Languages with higher token-to-word ratio are assigned larger proportions while others receive smaller proportions.

$$t_l^N = \frac{w_l^N}{\sum_{k \in L} w_k^N} \tag{3}$$

To avoid abrupt shifts in the sampling distribution from one iteration to the next, the target proportions are combined with the previous mixture using an exponential moving average. This results in an updated mixture computed as a weighted combination of the past and current targets, controlled by smoothing factor $\mu$, that determine aggressiveness of weight redistribution. Smaller $\mu$ leads to slower changes and preserving stability, whereas a larger $\mu$ allows faster adaptation. This mechanism ensures that mixture adjustments are gradual and stable, reducing the risk of over-correction.

$$m_l^N = (1 - \mu) \cdot m_l^{N-1} + \mu \cdot t_l^N \tag{4}$$

Once the updated mixture is computed for each language, it is scaled by the sampling budget to determine the actual number of characters to be allocated for each language. The value is rounded to the nearest integer and normalized to adjust for the small deviations caused by the rounding. This step finalizes how much training data each language will contribute in the next tokenizer iteration.

$$C_l^N = \text{round}(m_l^N \cdot T) \tag{5}$$

Together, these steps form a feedback driven optimization loop that adaptively updates data mixtures based on the tokenization performance. This ensures that under performing languages receive increased representation over time, while well performing languages are not destabilized. The entire process can be expressed in a single consolidated equation as given below:

$$m_l^N = (1 - \mu) \cdot m_l^{N-1} + \mu \cdot \left( \frac{\frac{f_l^N - f_{\text{best}}}{f_{\text{range}}^N} + \varepsilon}{\sum_{k \in L} \left( \frac{f_k^N - f_{\text{best}}}{f_{\text{range}}^N} + \varepsilon \right)} \right) \tag{6}$$

This formula is applied iteratively for each $N$, and $m_l^N$ is re-normalized if the sum deviates from 1.

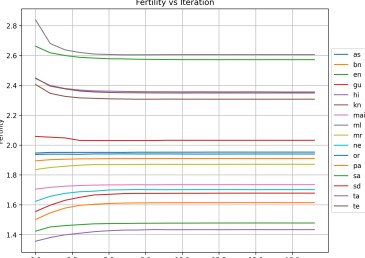

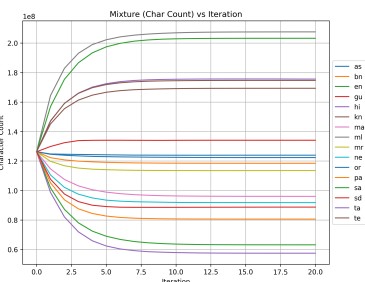

(a) Fertility Optimization across Iterations.

(b) Optimal mixture allocation.

Figure 1: Fertility and Mixture Allocation across Iterations for AdaptMix Tokenizer

If $f_{\text{range}}^N = 0$, then:

$$m_l^N = \frac{m_l^{N-1}}{\sum_{k \in L} m_k^{N-1}} \cdot T \tag{7}$$

To evaluate the effectiveness of the strategy, a series of controlled experiments was conducted. All tokenizers were trained using BPE with a vocabulary size of 128K and no pre-tokenization beyond optional byte-level splitting. The training data size was kept constant , augmented with a fixed code-math corpus to ensure coverage of technical symbols. We evaluated 4 data mixtures in total, shown in Figure 2. The adaptive algorithm began with from a uniform distribution and adjusted the sampling distribution iteratively based on the observed fertility for each language. Tokenizers were trained over 20 mixture-adjustment iterations, each involving full training, fertility analysis, and reweighting. The evolution of language wise fertility across iterations is shown in Figure 1.

To assess whether improvements in fertility translated to improved model performance, small language models were trained using each tokenizer variant. Each model was trained on the same dataset and initialization, using only the tokenizer as the variable component. We then evaluated each model's perplexity on a multilingual held-out test set to assess downstream performance.

## 4 RESULTS

### 4.1 VOCABULARY, BPE AND UNIGRAM

A comprehensive set of experiments was conducted to evaluate the impact of vocabulary size on tokenizer performance, with vocabulary sizes of 32K, 64K, 128K and 256K for both Byte-Pair Encoding(BPE) Sennrich et al. (2016) and Unigram Kudo (2018) algorithms. The results, presented in Table 1, highlight the token-to-word ratio of the key metric. Byte-Level tokenizers demonstrated better performance in terms of token-to-word ratio across multiple configuration of both BPE and

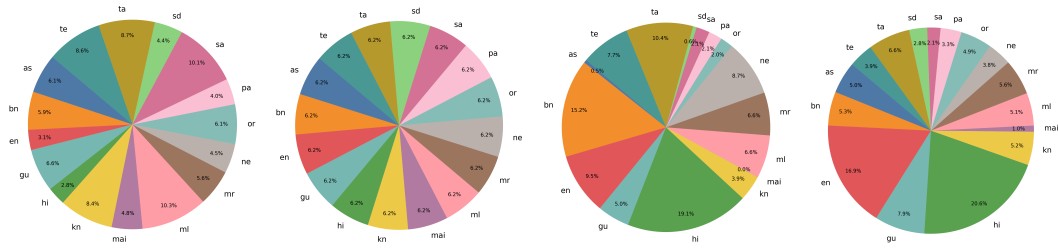

(a) AdaptMix Distribution    (b) Uniform Distribution    (c) Sangraha Distribution    (d) Skewed Distribution

Figure 2: **Data Mixture Comparison** 'AdaptMix' (Optimal) our proposed adaptive sampling based on tokenization difficulty. 'EnHiMix' (Skewed) biases the data toward English and Hindi. 'UniMix' (Uniform) applies uniform sampling across languages. 'SangrahaMix' reflects the distribution in the Sangraha dataset Khan et al. (2024). Language abbreviations used are listed in Appendix A.1

Table 1: Token-to-word ratio Comparison for different Vocabulary Size

| Language | BPE 32K | BPE 64K | BPE 128K | BPE 256K BL | Unigram 32K | Unigram 64K BL | Unigram 128K | Unigram 256K BL |
|---|---|---|---|---|---|---|---|---|
| Assamese | 2.15 | 1.75 | 1.5 | 1.37 | 2.51 | 2.31 | 2.27 | 2.27 |
| Bengali | 4.37 | 3.68 | 3.26 | 3.01 | 4.51 | 4.01 | 3.86 | 3.82 |
| English | 3.86 | 3.49 | 3.22 | 3.01 | 4.17 | 3.73 | 3.45 | 3.33 |
| Gujarati | 2.89 | 2.44 | 2.17 | 2 | 3.11 | 2.76 | 2.64 | 2.62 |
| Hindi | 2.14 | 1.94 | 1.83 | 1.78 | 2.34 | 2.2 | 2.15 | 2.15 |
| Kannada | 4.1 | 3.52 | 3.18 | 2.98 | 4.24 | 3.79 | 3.68 | 3.66 |
| Maithili | 2.53 | 2.18 | 1.98 | 1.85 | 2.82 | 2.6 | 2.53 | 2.51 |
| Malayalam | 2.88 | 2.47 | 2.23 | 2.09 | 3.12 | 2.85 | 2.77 | 2.76 |
| Marathi | 3.02 | 2.56 | 2.3 | 2.15 | 3.38 | 3.13 | 3.06 | 3.05 |
| Nepali | 2.93 | 2.52 | 2.27 | 2.13 | 3.21 | 2.93 | 2.84 | 2.82 |
| Odia | 3.82 | 3.26 | 2.94 | 2.75 | 3.96 | 3.58 | 3.47 | 3.45 |
| Punjabi | 2.6 | 2.27 | 2.06 | 1.92 | 2.88 | 2.72 | 2.68 | 2.68 |
| Sanskrit | 2.4 | 2.15 | 2 | 1.9 | 2.64 | 2.43 | 2.34 | 2.32 |
| Sindhi | 1.92 | 1.64 | 1.48 | 1.39 | 2.32 | 2.1 | 2.03 | 2.02 |
| Tamil | 2.79 | 2.44 | 2.22 | 2.09 | 3.05 | 2.8 | 2.71 | 2.69 |
| Telugu | 3.61 | 3.14 | 2.85 | 2.67 | 3.75 | 3.38 | 3.25 | 3.23 |
| **Average** | **3** | **2.59** | **2.34** | **2.19** | **3.25** | **2.96** | **2.86** | **2.84** |

*The table compares token-to-word ratio for different vocabulary size of 32K, 64K, 128K and 256K across multiple Indian languages. BL denotes byte level segmentation.*

Unigram algorithms. Among various sizes, the vocabulary size of 128K emerged as the most balanced configuration. It offers an effective tradeoff between token-to-word ratio and model efficiency, especially considering the inclusion of mathematical symbols, programming language tokes, and reserved special tokens. While the 256K vocabulary showed marginal improvement in token-to-word ratio, it effectively doubles the embedding matrix size, leading to significant overhead in memory consumption and model performance. Further analysis revealed that certain languages exhibited a persistently high token-to-word ratio even at a larger vocabulary size. This phenomenon was attributed to linguistic features such as *Sandhi Vibhajan* (Morphological Fusion), a morphological rule prevalent in many Indic languages, where multiple words are merged into a single compound word. Such language-specific phenomenon introduce challenges in achieving a low token-to-word ratio. While Unigram tokenization yields results that are only slightly inferior to BPE at a vocabulary size of 32K, its token-to-word ratio deteriorates with a large vocabulary size. The probabilistic nature of the unigram model encounters numerical instability resulting in NaN errors during training.

## 4.2 Pre-Tokenization

Pre-tokenization strategies significantly influence the efficiency and quality of the tokenizer by standardizing input text and reducing redundancy. We investigated multiple pre-tokenization methods, particularly focusing on the segmentation of diacritics common in Indic scripts. Two distinct strategies were evaluated: one strategy involved separating all diacritics, while the other selectively separated only a subset aimed at optimizing the token-to-word ratio. Empirical results, summarized in Table 2, reveal nuanced impacts of pre-tokenization strategies. Surprisingly, our experiments indicate that applying aggressive pre-tokenization consistently worsened the fertility scores across most

languages, contrary to our initial hypothesis. This finding suggests that excessive pre-tokenization can lead to unnecessary fragmentation, diminishing the overall token-to-word ratio.

However, evaluating the token-to-word ratio alone did not provide a comprehensive picture, and thus, assessing perplexity scores was also essential. To investigate this, we trained a 100M parameter model using each of the pre-tokenization strategies, ensuring that model configuration was consistent across experiments. All pre-tokenization strategies yielded substantially better perplexity scores than without pre-tokenization baseline, with clear variations observed across strategies in Table 3.

Table 2: Token-to-Word ratio comparison for different Pre-Tokenization Strategies

| Language | PT-0 BPE | PT-0 Unigram | PT-1 BPE | PT-1 Unigram | PT-2 BPE | PT-2 Unigram |
|---|---|---|---|---|---|---|
| Assamese | 1.88 | 3.21 | 2.27 | 2.84 | 3.11 | 3.69 |
| Bengali | 1.85 | 3.15 | 2.23 | 2.77 | 3.26 | 3.77 |
| English | 1.45 | 2.75 | 1.48 | 2.03 | 1.43 | 1.76 |
| Gujarati | 1.83 | 3.15 | 2.17 | 2.64 | 3.01 | 3.6 |
| Hindi | 1.35 | 2.66 | 1.83 | 2.15 | 2.58 | 2.88 |
| Kannada | 2.21 | 3.41 | 2.94 | 3.47 | 4.09 | 4.83 |
| Maithili | 1.71 | 2.87 | 2 | 2.34 | 2.57 | 2.91 |
| Malayalam | 2.72 | 3.76 | 3.26 | 3.86 | 4.89 | 5.9 |
| Marathi | 1.72 | 3.14 | 2.22 | 2.71 | 3.44 | 3.81 |
| Nepali | 1.65 | 3 | 1.98 | 2.53 | 3.15 | 3.61 |
| Odia | 1.87 | 3.3 | 2.3 | 3.06 | 3.27 | 4.04 |
| Punjabi | 1.65 | 3.14 | 2.06 | 2.68 | 2.64 | 3.26 |
| Sanskrit | 3.02 | 3.7 | 3.22 | 3.45 | 4.09 | 4.67 |
| Sindhi | 1.54 | 3.19 | 1.5 | 2.27 | 1.44 | 1.93 |
| Tamil | 2.16 | 3.41 | 2.85 | 3.25 | 4.43 | 5.28 |
| Telugu | 2.44 | 3.5 | 3.18 | 3.68 | 4.2 | 4.9 |
| **Average** | **1.94** | **3.21** | **2.34** | **2.86** | **3.23** | **3.8** |

Table 3: Perplexity Score Comparison for Different Pre-tokenization strategies

| Language | PT-0 BPE | PT-1 BPE | PT-2 BPE | PT-0 Unigram | PT-1 Unigram | PT-2 Unigram |
|---|---|---|---|---|---|---|
| Assamese | 94.56 | 40.55 | 59.62 | 39.87 | 32.75 | 39.84 |
| Bengali | 116.63 | 42.41 | 70.26 | 47.34 | 35.96 | 47.08 |
| English | 153.34 | 167.9 | 136.16 | 42.96 | 83.6 | 42.74 |
| Gujarati | 101.66 | 44.81 | 69.55 | 42.5 | 35.3 | 42.49 |
| Hindi | 97.12 | 36.17 | 54.68 | 35.34 | 31.99 | 35.43 |
| Kannada | 102.86 | 40.56 | 59.42 | 50.8 | 32.17 | 50.77 |
| Maithili | 124.97 | 50.4 | 77.77 | 45.09 | 41.68 | 45.2 |
| Malayalam | 92.21 | 39.15 | 61.54 | 50.83 | 30.94 | 50.92 |
| Marathi | 154.23 | 44.7 | 84.44 | 51.93 | 39.02 | 52.12 |
| Nepali | 139.38 | 48.23 | 86.75 | 52.86 | 41.64 | 52.88 |
| Odia | 93.14 | 40.14 | 63.61 | 40.66 | 32.19 | 40.76 |
| Punjabi | 81.88 | 41.03 | 54.06 | 33.73 | 32.36 | 33.76 |
| Sanskrit | 70.76 | 32.74 | 50.57 | 40.8 | 29.51 | 40.86 |
| Sindhi | 101.42 | 46.9 | 62.61 | 43.5 | 38.59 | 43.75 |
| Tamil | 107.17 | 35.9 | 61.5 | 49.68 | 29.07 | 49.81 |
| Telugu | 95.51 | 39.55 | 55.51 | 49.41 | 32.04 | 49.29 |
| **Average** | **107.93** | **69.25** | **49.45** | **44.83** | **44.86** | **37.43** |

*The table compares perplexity scores across different Indian languages using two tokenization algorithms: SentencePiece Byte Pair Encoding (BPE) and Unigram, under three distinct pre-tokenization strategies: PT-0 (Baseline, without pre-tokenization), PT-1 (Pre-tokenization of certain diacritics), and PT-2 (Pre-tokenization of all diacritics). Lower perplexity scores indicate better tokenization performance.*

Table 4: Token-to-word ratio Comparison Across Mixtures

| Language | **AdaptMix** | EnHiMix | UniMix | SangrahaMix |
|---|---|---|---|---|
| Assamese | 1.93 | 2.47 | 1.93 | 2.35 |
| Bengali | 1.90 | 2.22 | 1.89 | 1.74 |
| English | 1.47 | 1.27 | 1.42 | 1.39 |
| Gujarati | 2.03 | 8.60 | 2.05 | 2.20 |
| Hindi | 1.43 | 1.18 | 1.35 | 1.30 |
| Kannada | 2.30 | 2.92 | 2.40 | 2.56 |
| Maithili | 1.73 | 1.71 | 1.70 | 1.88 |
| Malayalam | 2.60 | 3.45 | 2.83 | 2.77 |
| Marathi | 1.87 | 1.86 | 1.83 | 1.78 |
| Nepali | 1.70 | 1.92 | 1.62 | 1.58 |
| Odia | 1.95 | 2.64 | 1.94 | 2.33 |
| Punjabi | 1.61 | 2.05 | 1.50 | 1.80 |
| Sanskrit | 2.57 | 2.97 | 2.66 | 2.91 |
| Sindhi | 1.67 | 1.70 | 1.55 | 1.73 |
| Tamil | 2.35 | 2.84 | 2.44 | 2.22 |
| Telugu | 2.34 | 2.73 | 2.44 | 2.39 |
| **Average** | **1.97** | **2.66** | **1.97** | **2.06** |

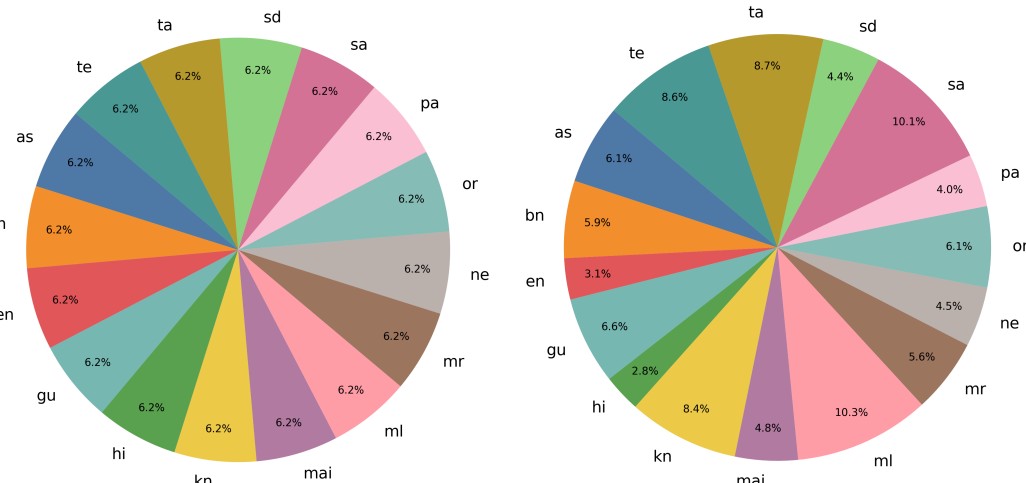

(a) Data distribution before applying AdaptMix      (b) Data distribution after applying AdaptMix

Figure 3: **Data Mixture Transition** *Left data distribution:* Initial Uniform distribution across languages. *Right data distribution:* After applying 'AdaptMix' algorithm to uniform distribution. Language abbreviations used are listed in Appendix A.1

### 4.3 ADAPTMIX: ADAPTIVE DATA MIXTURE

Results demonstrated consistent and interpretable trends; languages such as Sanskrit, Tamil, Malayalam, etc. initially exhibited higher fertility, but showed steady reductions across iterations until the optimal mixture was reached. At the same time, languages that started with low fertility, such as English, Maithili and Punjabi, retained stable performance, indicating that the optimization algorithm retained its efficiency.

As per Figure 3 it is evident that substantial cross-linguistic variation necessitates assigning appropriate weights to each language so that a balanced tokenizer attains an optimal token-to-word ratio across languages. Sanskrit and Dravidian languages (Kannada, Malayalam, Telugu, Tamil) require greater weights. We found their agglutination and longer orthographic word forms yield higher token rates than in other languages. Similar insight is shared by Saluja et al. (2019) for these languages.

To assess generalization, we experimented with varying vocabulary sizes between 16K and 128K. Despite the variation, the results showed consistent allocation patterns with deviations within 1–2%, suggesting robustness to vocabulary size.

Tokenizer trained using AdaptMix algorithm, with the optimal data mixture shown in 2a, achieved the lowest fertility score among all evaluated tokenizers across all 16 languages. To validate the effectiveness of the optimal mixture, Table 4 shows a comparison between 4 different data sampling strategies, while keeping tokenizer configuration same.

To assess the downstream impact of tokenization strategies, identical models were trained using each tokenizer variant and evaluated on a held out test set. All models shared the same configuration and training data, ensuring that the tokenizer was the only differing factor. The optimal mixture tokenizer achieved the lowest overall perplexity, with improvements in morphologically rich languages like Bengali, Malayalam, Oriya, and Tamil, while maintaining strong performance on high resource languages like English and Hindi. Notably, the English/Hindi heavy tokenizer excels on its focus languages but performs poorly on others. Random and uniform mixtures show inconsistent results due to a lack of adaptive balancing. These findings reinforce the earlier analysis on fertility and parity, demonstrating that improvements in tokenization quality translate to downstream performance benefits.

Table 5: Token-to-word ratio Comparison Across state-of-the-art Tokenizers

| Language | AdaptMix | Qwen | LLaMA | Nemotron Mistral | Nemotron Mini | Sarvam-M | DeepSeekv3 | Gemma 27B | Airavata |
|---|---|---|---|---|---|---|---|---|---|
| Assamese | **1.93** | 7.18 | 8.06 | 4.24 | 4.58 | 4.24 | 3.59 | 2.68 | 8.94 |
| Bengali | 1.90 | 6.92 | 7.85 | 2.93 | 2.65 | 2.93 | 2.89 | **1.74** | 8.20 |
| English | 1.47 | 1.36 | 1.35 | 1.37 | 1.35 | 1.37 | 1.33 | **1.35** | 1.58 |
| Gujarati | **2.03** | 8.53 | 9.54 | 3.59 | 15.17 | 3.59 | 4.84 | 2.39 | 14.14 |
| Hindi | 1.43 | 4.66 | 2.65 | 1.97 | 1.77 | 1.97 | 2.92 | **1.38** | 1.80 |
| Kannada | **2.30** | 11.08 | 13.81 | 3.82 | 4.02 | 3.82 | 5.83 | 3.15 | 19.35 |
| Maithili | **1.73** | 4.67 | 2.85 | 2.53 | 2.28 | 2.53 | 3.28 | 1.90 | 2.45 |
| Malayalam | **2.60** | 13.30 | 16.00 | 4.88 | 4.71 | 4.88 | 7.83 | 3.39 | 11.38 |
| Marathi | **1.87** | 6.46 | 3.86 | 3.14 | 2.62 | 3.14 | 4.15 | 1.94 | 3.31 |
| Nepali | **1.70** | 6.28 | 3.61 | 3.04 | 2.32 | 3.04 | 4.07 | 2.06 | 3.05 |
| Oriya | 1.95 | 12.92 | 15.91 | **1.71** | 17.24 | 17.23 | 7.26 | 4.60 | 17.29 |
| Punjabi | **1.61** | 7.39 | 7.88 | 3.12 | 12.70 | 3.12 | 4.51 | 2.74 | 10.84 |
| Sanskrit | **2.57** | 8.00 | 4.75 | 4.26 | 4.32 | 4.26 | 4.95 | 3.36 | 4.66 |
| Sindhi | **1.67** | 3.09 | 2.99 | 2.65 | 2.83 | 2.65 | 2.98 | 2.14 | 5.04 |
| Tamil | **2.35** | 9.75 | 11.89 | 3.71 | 3.57 | 3.71 | 4.88 | 2.42 | 10.50 |
| Telugu | **2.34** | 11.45 | 13.30 | 3.90 | 3.77 | 3.90 | 5.99 | 2.93 | 19.51 |
| **Average** | **1.97** | **7.69** | **7.89** | **3.18** | **5.37** | **4.15** | **4.46** | **2.51** | **8.88** |

Table 5 shows AdaptMix performs better across the state-of-the-art tokenizers in terms of the token-to-word ratio properly balancing the ratio of different languages.

## 5 CONCLUSION

We presented a comprehensive analysis of multilingual tokenizer strategies and demonstrated that any optimal vocabulary size of 128K effectively balances tokenization efficiency and computational constraints, outperforming both smaller or larger vocabularies. Furthermore, various pre-tokenization methods improves models performance, despite a slight increase in the token-to-word ratio. In our experiment we varied the mixture weights across Indic languages to study their impact on tokenizer and model performance. Our proposed **AdaptMix** algorithm dynamically optimized multilingual training data composition for all languages, significantly reducing disparity of token-word-ratio across languages. Collectively these contribution underline tokenizer as a fundamental component on par with model architecture and training objectives in building scalable and efficient multilingual language models.

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

## A  APPENDIX

### A.1  ABBREVIATIONS

| Abbreviation | Language |
|---|---|
| as | Assamese |
| bn | Bengali |
| en | English |
| gu | Gujarati |
| hi | Hindi |
| kn | Kannada |
| mai | Maithili |
| ml | Malayalam |
| mr | Marathi |
| ne | Nepali |
| or | Odia |
| pa | Punjabi |
| sd | Sindhi |
| sa | Sanskrit |
| ta | Tamil |
| te | Telugu |

### A.2  TOKEN TO WORD RATIO

Token to Word Ratio is also commonly known as Fertility score. It is defined as,

$$\text{Token-to-Word Ratio} = \frac{\text{Total Number of Tokens}}{\text{Total Number of Words}} \tag{8}$$

### A.3 MODEL TRAINING DETAILS

All models were trained from scratch using causal decoder transformer Vaswani et al. (2017b) architecture with 16 layers and a hidden size of 512, resulting in approximately 100M parameters. Each model used a vocabulary size of 128k, based on the tokenizer being evaluated. The optimizer used was AdamW Loshchilov & Hutter (2017) with a learning rate of 3e-4, cosine learning rate decay Loshchilov & Hutter (2016) and the weight decay set to 0.1. Training was performed using BF16 precision on a single node with 2 GPUs, using Fully Sharded Data Parallelism (FSDP) Zhao et al. (2023) for efficient memory and compute scaling.

### A.4 PROBLEMS WITH WEIGHTED DATA MIXTURE OPTIMIZATION

Prior to scaling optimization algorithm to 16 languages, preliminary experiments were conducted on a subset of four languages: English, Punjabi, Malayalam and Sanskrit. The subset was chosen to reflect a range of token-to-word ratio behaviors as these languages have unique character sets and grammatical rules. English and Punjabi generally perform well over default mixtures, whereas Malayalam and Sanskrit are observed to exhibit higher token-to-word ratio. These small scale experiments helped to mold core algorithm without compute over-utilization.

However, during the early stage experiments, we observed unintended behavior in the way token-to-word ratio deficits were calculated. Initially, the best token-to-word ratio was defined dynamically as the lowest token-to-word ratio among all languages in each iteration. While this allowed the algorithm to adaptively update the mixture, it introduced a problematic patterns; instead of increasing the proportion of under-performing languages, the optimizer began decreasing the proportion of well performing ones, as observed in Figure 4. This occurred because Sanskrit and Malayalam could not realistically reach the same tokenization efficiency as English or Punjabi within the same vocabulary size. As a result, the algorithm minimized the overall deficit by degrading the performance of already efficient languages instead of improving under-performing ones.

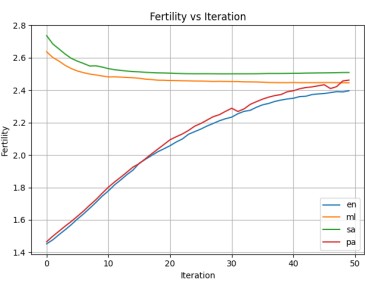 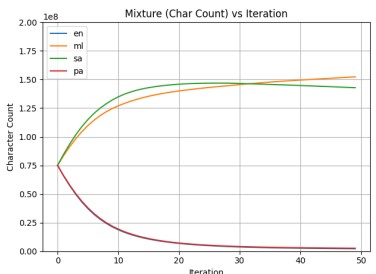

(a) Weighted Fertility across Iterations.          (b) Weighted Mixture allocation.

Figure 4: Fertility and Mixture Allocation across Iterations for Weighted Algorithm

This problem was resolved using our novel AdaptMix Algorithm on the same set of dataset and configuration as shown in 5.

### A.5 PARITY CALCULATION

In addition to evaluating fertility, Table 6 shows tests conducted on Parity Petrov et al. (2023), which quantifies cross lingual fairness and bias in tokenization. Across the 16 Indian languages, the optimal data mixture consistently outperformed state of the art open source model tokenizers like Qwen, LLama, DeepSeek, and Gemma in achieving parity with English.

### A.6 APDATMIX PSEUDO CODE

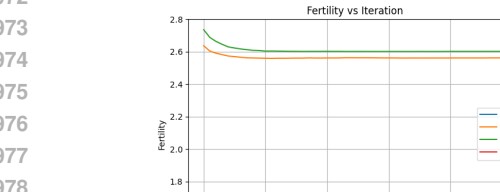

(a) AdaptMix Fertility across Iterations.

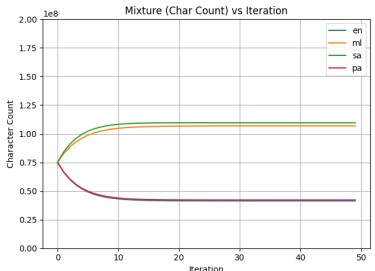

(b) AdaptMix Mixture allocation.

Figure 5: Fertility and Mixture Allocation across Iterations for AdaptMix Algorithm

Table 6: Parity Comparison Across state-of-art Tokenizers

| Language | **AdaptMix** | Qwen | LLaMA | Nemotron Mistral | Nemotron Mini | Sarvam-M | DeepSeek v3 | Gemma 27B |
|---|---|---|---|---|---|---|---|---|
| Assamese | 1.31 | 5.27 | 5.97 | 3.09 | 3.39 | 3.09 | 2.69 | 1.98 |
| Bengali | 1.29 | 5.08 | 5.81 | 2.13 | 1.96 | 2.13 | 2.17 | 1.28 |
| English | 1.00 | 1.00 | 1.00 | 1.00 | 1.00 | 1.00 | 1.00 | 1.00 |
| Gujarati | 1.37 | 6.27 | 7.06 | 2.62 | 11.23 | 2.62 | 3.63 | 1.77 |
| Hindi | 0.96 | 3.42 | 1.96 | 1.43 | 1.31 | 1.43 | 2.19 | 1.02 |
| Kannada | 1.56 | 8.14 | 10.22 | 2.78 | 2.97 | 2.78 | 4.38 | 2.33 |
| Maithili | 1.17 | 3.43 | 2.11 | 1.84 | 1.68 | 1.84 | 2.46 | 1.40 |
| Malayalam | 1.76 | 9.77 | 11.85 | 3.56 | 3.48 | 3.56 | 5.88 | 2.51 |
| Marathi | 1.26 | 4.75 | 2.85 | 2.29 | 1.94 | 2.29 | 3.12 | 1.43 |
| Nepali | 1.15 | 4.61 | 2.67 | 2.21 | 1.71 | 2.21 | 3.06 | 1.52 |
| Oriya | 1.32 | 9.50 | 11.78 | 12.57 | 12.77 | 12.57 | 5.45 | 3.40 |
| Punjabi | 1.09 | 5.43 | 5.83 | 2.27 | 9.40 | 2.27 | 3.39 | 2.02 |
| Sanskrit | 1.74 | 5.88 | 3.51 | 3.10 | 3.20 | 3.10 | 3.72 | 2.48 |
| Sindhi | 1.13 | 2.27 | 2.21 | 1.93 | 2.09 | 1.93 | 2.24 | 1.58 |
| Tamil | 1.59 | 7.16 | 8.80 | 2.70 | 2.64 | 2.70 | 3.66 | 1.79 |
| Telugu | 1.58 | 8.41 | 9.85 | 2.84 | 2.79 | 2.84 | 4.50 | 2.17 |

---

**Algorithm 1** Calculate Mixture

---

**Require:** Current mixture $M_t$, token-to-word ratio/fertility scores $F$, learning rate $\lambda$, constant $\epsilon$
**Ensure:** New mixture $M_{t+1}$
1: $total\_chars \leftarrow \sum_i current\_mixture[l_i]$

2: $best\_fertility \leftarrow \min(F.values())$
3: $worst\_fertility \leftarrow \max(F.values())$
4: $fertility\_range \leftarrow worst\_fertility - best\_fertility$
5: **for** each language $l_i$ **do**
6:     **if** $fertility\_range > 0$ **then**
7:         $scaled\_deficit[l_i] \leftarrow (F[l_i] - best\_fertility)/fertility\_range$
8:     **else**
9:         $scaled\_deficit[l_i] \leftarrow 0$
10:     **end if**
11:     $deficit\_weight[l_i] \leftarrow scaled\_deficit[l_i] + \epsilon$
12: **end for**

13: $total\_deficit \leftarrow \sum_i deficit\_weight[l_i]$
14: **for** each language $l_i$ **do**
15:     $deficit\_target[l_i] \leftarrow deficit\_weight[l_i]/total\_deficit$
16:     $new\_prop[l_i] \leftarrow current\_prop[l_i](1 - \lambda) + deficit\_target[l_i]\lambda$
17:     $current\_prop[l_i] \leftarrow current\_mixture[l_i]/total\_chars$
18:     $new\_mixture[l_i] \leftarrow \lfloor new\_prop[l_i] \times total\_chars \rfloor$
19: **end for**
20: **return** $new\_mixture$

