# OpenReview forum: "The Art of Breaking Words: Rethinking Multilingual Tokenizer Design"
_ICLR.cc/2026/Conference — Submitted to ICLR 2026_

### Official Review · Reviewer_Zeip · 2025-10-22

**Soundness:** 2
**Presentation:** 1
**Contribution:** 1
**Rating:** 0
**Confidence:** 5

**Summary:**

The paper provides an analysis of how different language mixtures affect the tokenizer token-to-word ratio.

**Strengths:**

Relatively comprehensive report on designing a dataset mixture for training a tokenizer for Indic languages.

**Weaknesses:**

1. Overall no particular novelty or contribution. Upweighting lower-resourced languages in the design of tokenizers has long been used. See (Arivazhagan et al., 2019, Costa-jussà et al., 2022). While the specific strategy might be different, the fundamental idea is the same.
2. There is very limited review of the literature on tokenizer design and especially on prior approaches for improving the data mixtures for training multilingual tokenizers.
3. The paper focuses on token-to-word ratios which are not fully informative for the real use of language models as different languages may use different quantities of words to express the same content. Furthermore, dividing sentences into words is not trivial for some languages (e.g., Japanese, Chinese) making the work not applicable to them.
4. Not described how perplexity is calculated. As the models use different tokenizers, the fertility of the tokenizer itself can also affect perplexity.
5. Perplexity is evaluated on a 100M parameter model but no further details about the model are provided.
6. The authors claim they have made a roughly 100B token synthetic dataset that then was translated into 15 languages. This implies that they have trained on about 1.6T tokens which would typically be expected for a model of 10x the size. The only details about this dataset provided are “using open source and filtered for quality” so not at all clear exactly what it is and how it was made.
7. Overall reads more like a technical report than a scientific paper: it provides details on how the authors built and designed a specific instance of a tokenizer but not much scientific or transferable insight.

Massively Multilingual Neural Machine Translation in the Wild: Findings and Challenges, Arivazhagan et al., 2019

No Language Left Behind: Scaling Human-Centered Machine Translation, Costa-jussà et al., 2022

Minor:
- Mixing up the \citet and \citep LaTeX commands.
- Line 251: Extra space before comma.

**Questions:**

See the Weaknesses section.

---

### Official Review · Reviewer_TKv5 · 2025-10-29

**Soundness:** 2
**Presentation:** 3
**Contribution:** 3
**Rating:** 4
**Confidence:** 4

**Summary:**

The authors primarily address the issue of high token-to-word ratios in multilingual tokenizers, particularly for Indic scripts. Specifically, they first investigate how vocabulary size and pre-tokenization rules affect the token-to-word ratio. Then, they propose AdaptMix, an adaptive data composition strategy designed to balance multilingual data in tokenizer training and thereby reduce the token-to-word ratio.

**Strengths:**

1.The authors constructed a dataset covering 16 Indian languages.

2.The paper is well-structured and clearly written; the proposed method is introduced in a concise and easy-to-understand manner.

**Weaknesses:**

1.The proposed AdaptMix method requires multiple iterations, which makes it more computationally expensive compared to other tokenizer training approaches.

2.The authors focus primarily on reducing the token-to-word ratio. Although this ratio is indeed an important indicator for tokenizer efficiency, a lower ratio does not necessarily guarantee better performance for large language models. It raises the question of whether reducing the ratio might compromise the semantic representation of morphologically complex languages, thereby affecting the overall performance of LLMs. In the paper, only Table 3 presents a comparison of perplexity (PPL) under different pre-tokenization strategies, but there are no further results demonstrating how the proposed method impacts model performance beyond tokenization efficiency.

**Questions:**

1.Some prior studies generally assume that larger models require larger vocabularies. Therefore, should the authors’ conclusion regarding the relationship between vocabulary size and the token-to-word ratio take into account the potential influence of model size?

2.Could the authors provide a comparison of time complexity of AdaptMix or actual runtime performance against other tokenizers?

---

### Official Review · Reviewer_GAb3 · 2025-11-01

**Soundness:** 2
**Presentation:** 1
**Contribution:** 2
**Rating:** 2
**Confidence:** 4

**Summary:**

This paper systematically examines how vocabulary size, pre-tokenization rules, and training-corpus composition affect token-to-word efficiency and model quality, and introduces a data-composition algorithm, AdaptMix, which lowers the average tokens-per-word ratio—particularly for multilingual Indic scripts.

**Strengths:**

S1: The authors focus on non‑Latin scripts and undertake foundational research on complex, understudied multilingual Indic language models, thereby providing a valuable basis for future work in the field.

S2: The authors provide a detailed analysis of tokenizer design—examining vocabulary size, pre‑tokenization rules, and data composition methods—which facilitates a multi‑level understanding of the proposed method’s effectiveness in low‑resource, morphologically complex language scenarios.

**Weaknesses:**

W1: The authors place excessive emphasis on the token‑to‑word metric (i.e., vocabulary compression rate). Prior work has shown that higher compression is not necessarily better; excessively high compression can degrade generalization, especially when transferring to new corpora. Therefore, the paper's strong emphasis on this single ratio is unjustified.

W2: In Section 4.1 the authors should adopt a more scientific and systematic criterion for selecting vocabulary size to quantify the trade‑off between decoding latency and the token‑to‑word ratio. As presented, the choice of sizes appears subjective—for example, why not select values in the 128K–256K range?

W3: In Table 4, i.e., one of the most important experiment results, the average token‑to‑word ratio for AdaptMix (1.97) is identical to that of UniMix (1.97, a uniform distribution). This result calls into question the practical advantage of the more complex AdaptMix algorithm over a simple baseline.

W4: The paper appears to consider only the token‑to‑word ratio and perplexity metrics, which leads to overly simplistic conclusions. The study should include deeper analyses — for example, additional case studies, broader baseline comparisons, and more comprehensive evaluation/diagnostic experiments.

W5: The authors should develop a more systematic and standardized analytical framework. The current analysis appear shallow and the derived insights are limited; the work reads more like a simple technical report than a comprehensive study.

W6: The mathematical notation and formulas require clearer exposition. For instance, in $f_{range}^{N}$ the meanings of "range" and of the superscript N should be explicitly defined and motivated.

**Questions:**

Q1: The paper states that a model trained with the AdaptMix tokenizer "achieved the lowest overall perplexity". Would it be possible to provide the table of perplexity scores comparing the models trained using the four different data mixtures (AdaptMix, UniMix, SangrahaMix, EnHiMix) from Section 4.3?

---

> ### Author Response · Authors · 2025-11-28
>
> W1: The Paper “BPE-Dropout: Simple and Effective Subword Regularization” demonstrates the regularization technique which randomly skips the merges which in a way randomly segments the words hence providing a robustness and generalization when training the model. We believe even if the ratio is 1 due to this ingenious approach introduced in this paper it prevents degradation of generalization.
>
> W2: The commonly adopted vocabulary sizes for models are around 32K, 128K and 256K by different models. Though multiple values can be taken from the range between 128K and 256K, but since models like qwen are utilizing “151642” as vocabulary size, it provides a very small difference of change in vocabulary and won’t provide detailed difference in fertility score for the amount of languages taken.
>
> W3: Although average fertility of UniMix and AdaptMix matches 1.97, this aggregation hides language specific improvements. As per table 4, the average fertility of Unimix(Uniform Mixture) and AdaptMix(Our approach) are same but, for languages having fertility more than 2, adaptmix was able to significantly reduce the fertility. Example: For Unimix sanskrit has 2.66 fertility and in AdaptMix has 2.57, which is a big improvement.
>
> W6: We will update the mathematical notation for more clearer notations.

---

### Official Review · Reviewer_jX5s · 2025-11-01

**Soundness:** 2
**Presentation:** 2
**Contribution:** 2
**Rating:** 4
**Confidence:** 3

**Summary:**

This paper presents a systematic study of multilingual tokenizer design, focusing on Indic languages with diverse scripts and orthographic systems. The authors introduce AdaptMix, an adaptive data mixture algorithm that dynamically adjusts language sampling based on token-to-word ratios (termed fertility). The work aims to reduce token fragmentation and achieve balanced tokenization efficiency across languages. Through extensive experiments covering vocabulary size, pre-tokenization strategies, and data mixture policies, the proposed tokenizer achieves a reported 6% improvement in average token-to-word ratio and over 40% gains compared to existing multilingual Indic models. The study concludes that tokenization should be treated as a critical factor in multilingual LLM design.

**Strengths:**

1. The paper is backed by a large-scale empirical study across 16 Indian languages and multiple domains (code, math, text). The experimental scope is impressive, covering both vocabulary scaling and pre-tokenization.
2. The iterative reweighting algorithm based on tokenization fertility is elegant and addresses a real gap in multilingual tokenizer design, i.e., unbalanced sampling that harms low-resource, morphologically complex languages.
3. The authors demonstrate deep awareness of Indic linguistic phenomena (e.g., ligatures, Sandhi, diacritics), which grounds the technical design in solid linguistic reasoning.
4. Tables and figures show consistent trends across multiple configurations. The fertility-based analysis provides a clear and interpretable measure of tokenizer efficiency.

**Weaknesses:**

1. Although perplexity is reported for small models, there are no large-scale experiments on full LLMs (e.g., GPT, LLaMA, Qwen) to confirm that tokenization improvements translate into stronger language modeling or instruction-following performance.
2. The AdaptMix algorithm is empirically effective but lacks theoretical discussion, convergence properties, relation to distributionally robust optimization, or statistical guarantees of balanced fertility.
3. While the focus on Indic scripts is justified, it would be helpful to see how AdaptMix generalizes to other multilingual families (e.g., Cyrillic, Arabic, or East Asian).
4. The paper could benefit from a clearer statement of contributions and notation consistency. Some definitions (like fertility normalization) are buried deep in the method section, making it harder to follow.

**Questions:**

1. Can the authors provide any evidence that AdaptMix-trained tokenizers improve large model performance (e.g., fine-tuning LLaMA or Gemma on Indic datasets)?
2. How stable is the AdaptMix iteration process? does it converge quickly, and is it sensitive to the smoothing factor μ?
3. Is there any theoretical intuition for why fertility balancing improves cross-lingual generalization beyond empirical correlation?
4. Will the authors release tokenizer code, vocabulary files, and mixture statistics to enable replication?

---

> ### Author Response · Authors · 2025-11-21
>
> Thank you for your thoughtful comment and time.
>
> Responses for the individual questions are given below.
>
> 1. We did not include downstream large-model fine-tuning experiments in this work, as the fine-tuning benchmarking is extremely broad, spanning parameter scales, fine tuning, tasks, and languages which makes any single downstream evaluation inherently incomplete and potentially misleading. For this reason, we restrict our scope to tokenizer-level and pretraining evaluations.
>
> 2. The AdaptMix iteration is empirically stable. In our experiments, the mixture distribution converges within roughly 5–7 iterations across languages. We also observe that convergence behavior is not strongly sensitive to the smoothing factor μ.]
>
> 3. We do not explicitly claim cross-lingual model generalization improvements in the paper. Our focus is on cross-lingual tokenizer fairness, which is expected to indirectly benefit multilingual models. The intuition is that fertility balancing prevents high-resource or low-complexity scripts from dominating the shared vocabulary, thereby giving morphologically richer scripts a proportional representation.
> Concretely, Figures 1a and 1b show that Malayalam fertility improves substantially from 2.8 to  2.6 after AdaptMix rebalancing. Malayalam has a much larger effective character inventory (14 vowels, 52 consonants, plus numerous vowel consonant combinations/diacritics) compared to English, which has only 26 base letters. With a larger initial symbol set, subword frequency is spread thinner, so these languages are more likely to be fragmented. If the shared vocabulary is learned from skewed data, high-frequency English subwords disproportionately occupy capacity, reducing representation for more complex scripts. Fertility balancing counteracts this by aligning vocabulary allocation with script/morphological complexity, improving fairness and potentially supporting more equitable multilingual modeling.
>
> 4. Yes. We will release the tokenizer training code, the final vocabulary files, and the AdaptMix mixture.

---

### Meta-Review · Area_Chair_UAFg · 2026-01-07

**Summary:**

Reviewers agreed that the paper aims to address an important issue in multilingual LLMs, namely token fragmentation and imbalance. Main concerns raised by reviewers:

1. The core contribution (reweighting based on fertility) is incremental relative to prior multilingual mixture balancing work, with limited discussion and comparison to similar prior work.
2. The evaluation was considered weak and incomplete, relying heavily on fertility and limited perplexity results from small models, without convincing evidence that tokenizer improvements translate into stronger downstream LLM performance.
3. Over-optimizing token-to-word ratio alone may be misleading, as higher compression does not necessarily imply better generalization or semantic representation.
4. Clarity and rigor issues, including unclear notation, missing evaluation details (perplexity computation, model description), subjective vocabulary-size choices, and lack of runtime or complexity analysis.

While the authors provided clarifications and acknowledged some limitations, I concur with the reviewers that the central concern about downstream impact and novelty is not resolved.

**Reviewer Concerns:**

1. The authors clarified that AdaptMix converges quickly and is empirically stable.
2. They committed to releasing tokenizer code, vocabularies, and mixture statistics, improving reproducibility.
3. They acknowledged notation issues and promised clarification in a revision.

Issues still remaining:

1. The paper still lacks convincing evidence that the approach improves model performance on downstream tasks.
2. The reliance on fertility as the primary metric remains insufficiently justified.
3. Novelty relative to prior multilingual mixture and tokenizer work is still unclear.

**Reviewer Scores:**

While no reviewer engaged in discussions, I do not believe they will have raised their scores since major issues they raised were not completely addressed.

---

### Decision · Program_Chairs · 2026-01-26

Reject